# Cas9-specific immune responses compromise local and systemic AAV CRISPR therapy in multiple dystrophic canine models

Chady H. Hakim [ID] [1,2], Sandeep R. P. Kumar[3,4,19], Dennis O. Pérez-López[1,19], Nalinda B. Wasala[1], Dong Zhang [ID] [5,6,7], Yongping Yue[1], James Teixeira[1], Xiufang Pan[1], Keqing Zhang[1], Emily D. Million[1], Christopher E. Nelson [ID] [8,9], Samantha Metzger[1], Jin Han[1], Jacqueline A. Louderman[1], Florian Schmidt [ID] [10,11,12], Feng Feng [ID] [1], Dirk Grimm [ID] [10,11,12], Bruce F. Smith [ID] [13,14], Gang Yao[15], N. Nora Yang [ID] [2], Charles A. Gersbach [ID] [8,9,16], Shi-jie Chen[5,6,7], Roland W. Herzog [ID] [3,4] & Dongsheng Duan [ID] [1,15,17,18 ✉]

Adeno-associated virus (AAV)-mediated CRISPR-Cas9 editing holds promise to treat many diseases. The immune response to bacterial-derived Cas9 has been speculated as a hurdle for AAV-CRISPR therapy. However, immunological consequences of AAV-mediated Cas9 expression have thus far not been thoroughly investigated in large mammals. We evaluate Cas9-specific immune responses in canine models of Duchenne muscular dystrophy (DMD) following intramuscular and intravenous AAV-CRISPR therapy. Treatment results initially in robust dystrophin restoration in affected dogs but also induces muscle inflammation, and Cas9-specific humoral and cytotoxic T-lymphocyte (CTL) responses that are not prevented by the muscle-specific promoter and transient prednisolone immune suppression. In normal dogs, AAV-mediated Cas9 expression induces similar, though milder, immune responses. In contrast, other therapeutic (micro-dystrophin and SERCA2a) and reporter (alkaline phosphatase, AP) vectors result in persistent expression without inducing muscle inflammation. Our results suggest Cas9 immunity may represent a critical barrier for AAV-CRISPR therapy in large mammals.

[1] Department of Molecular Microbiology and Immunology, The University of Missouri, Columbia, MO, USA. [2] National Center for Advancing Translational Sciences, NIH, Rockville, MD, USA. [3] Department of Pediatrics, Indiana University, Indianapolis, IN, USA. [4] Herman B Wells Center for Pediatric Research, Indiana University, Indianapolis, IN, USA. [5] Department of Physics, The University of Missouri, Columbia, MO, USA. [6] Department of Biochemistry, The University of Missouri, Columbia, MO, USA. [7] Institute for Data Science and Informatics, The University of Missouri, Columbia, MO, USA. [8] Department of Biomedical Engineering, Duke University, Durham, NC, USA. [9] Center for Advanced Genomic Technologies Biology, Duke University, Durham, NC, USA. [10] Department of Infectious Diseases/Virology, University of Heidelberg, Heidelberg, Germany. [11] Cluster of Excellence CellNetworks, University of Heidelberg, Heidelberg, Germany. [12] BioQuant, University of Heidelberg, Heidelberg, Germany. [13] Department of Pathobiology, Auburn University, Auburn, AL, USA. [14] Scott-Ritchey Research Center, Auburn University, Auburn, AL, USA. [15] Department of Biomedical, Biological & Chemical Engineering, The University of Missouri, Columbia, MO, USA. [16] Department of Surgery, Duke University Medical Center, Durham, NC, USA. [17] Department of Neurology, The University of Missouri, Columbia, MO, USA. [18] Department of Biomedical Sciences, The University of Missouri, Columbia, MO, USA. [19]These authors contributed equally: Sandeep R. P. Kumar, Dennis O. Pérez-López. ✉email: duand@missouri.edu

CRISPR editing is an appealing strategy to repair disease-causing mutations in the human genome[1,2]. Ex vivo CRISPR therapy has revealed favorable outcomes in monogenic blood diseases, and AAV-mediated in vivo CRISPR therapy has also been initiated to treat an inherited retinal disease in human patients[3,4]. Despite these advances, Cas9 immunity remains poorly understood[5–7]. Natural immunity to Cas9 has been documented in humans[8–10]. Cas9-specific humoral and cellular responses have also been observed in mice injected with AAV Cas9 vectors, albeit it is unclear if a cytotoxic T cell response may occur[11–14].

Duchenne muscular dystrophy (DMD) is a lethal muscle disease caused by null mutations in the dystrophin gene[15]. We have achieved persistent dystrophin restoration and muscle and heart function rescue with AAV-CRISPR in the mdx mouse model[16,17]. Others have demonstrated AAV-CRISPR editing in ΔE50 canine and Δ52 swine DMD models[18,19].

Here we apply local and systemic AAV CRISPR therapy in three different canine DMD models. We show that AAV-mediated Cas9 expression induces humoral and cytotoxic T-lymphocyte (CTL) responses in normal and dystrophic canines. Cas9-specific CTL eliminates CRISPR-rescued dystrophin in affected dogs. We further show that the muscle-specific promoter and prednisolone transient immune suppression are insufficient to mitigate Cas9 immunity.

## Results and discussion

**gRNA design and screening**. To comprehensively evaluate AAV-mediated CRISPR therapy in the canine DMD model, we performed the study in golden retriever muscular dystrophy (GRMD), Welsh corgi muscular dystrophy (WCMD), and Labrador retriever muscular dystrophy (LRMD) dogs[20,21]. GRMD carries a point mutation in intron 6, which can be corrected using SaCas9 (Staphylococcus aureus-derived Cas9) and two gRNAs targeting mutation-flanking introns. WCMD and LRMD carry insertions in intron 13 and 19, respectively. Insertion leads to pseudoexons that disrupt the open reading frame. LRMD and WCMD can be treated with SpCas9 (Streptococcus pyogenes-derived Cas9) and a single gRNA targeting the splice acceptor (SA) and/or exonic splicing enhancer (ESE).

We designed a total of 12 pairs of gRNAs for GRMD, 6 SA/ESE-targeting gRNAs for LRMD, and 10 SA/ESE-targeting gRNAs for WCMD and screened these gRNAs in vitro (Supplementary Figs. 1–3 and Supplementary Tables 1–6). We then packaged the Cas9 gene expression cassette and the best gRNAs in AAV8 and screened them in vivo in dystrophic dogs by intramuscular injection. A total of 10 Cas9/gRNA vector combinations were evaluated (Supplementary Figs. 4 and 5a). Successful dystrophin restoration was achieved in all three models though efficiency varied for different Cas9/gRNA vector combinations (Supplementary Fig. 4b). The best Cas9/gRNA vectors were used in subsequent studies (Supplementary Fig. 4b, c).

**Pre-existing humoral immunity to Cas9 in canines**. To study dystrophin rescue and Cas9 immune responses, we first profiled the pre-existing Cas9 antibody in dogs. High levels of anti-Cas9 IgG were detected in adult dogs, while newborn puppies only had moderate levels of the maternal-derived Cas9 antibody, which dropped to barely detectable levels between 2 and 6 weeks of age (Supplementary Fig. 6a–e).

**AAV CRISPR therapy-induced CTL clears rescued dystrophin**. Next, we tested CRISPR therapy in affected dogs. To minimize the potential immune reaction, we expressed the Cas9 gene from the muscle-specific creatine kinase 8 (CK8) promoter and applied high-dose prednisolone immune suppression in all Cas9 vector-injected dogs (Fig. 1a and Supplementary Fig. 6f)[18,22–24]. To study local editing, we delivered CRISPR vectors to a 1-month-old LRMD dog (Fig. 1b–i and Supplementary Fig. 5b). Injected muscle was harvested into four blocks at 6-weeks post-injection, and distant muscles were harvested as the non-injected control (Fig. 1b–g). Similar to the ΔE50 model study[18], robust dystrophin rescue was observed by immunostaining and western blot in a vector quantity-dependent manner (Fig. 1c, d, g and Supplementary Fig 5b). However, unlike in the ΔE50 model study[18], we detected abundant CD4+ and CD8+ T-cell infiltration and vector quantity-dependent Cas9 expression and muscle cytokine elevation (Fig. 1b, d–g). Serum Cas9 antibody and interferon (INF)-γ ELISpot assay on peripheral blood mononuclear cells (PBMCs) revealed Cas9-specific immune responses (Fig. 1h, i). Similarly, abundant T-cell infiltration was observed in a CRISPR-treated adult GRMD dog (Supplementary Fig. 7a).

To determine whether this immune response eliminates CRISPR-rescued dystrophin, we co-injected CRISPR vectors and AAV.RSV.AP to two 44-month-old WCMD dogs with pre-existing Cas9 immunity (Fig. 1j–p, r and Supplementary Figs. 5c and 7b, c). Dystrophin and AP expression was detected at 3 weeks post-injection but substantially reduced at 6 weeks post-injection (Fig. 1j, n, k and Supplementary Fig. 5c). The loss of expression was accompanied by abundant CD4+ and CD8+ T-cell infiltration, vector genome reduction, and muscle cytokine elevation (Fig. 1j, i, r and Supplementary Fig. 7c). Serum Cas9 antibody and PBMC ELISpot indicated Cas9-specific immune responses (Fig. 1o, p and Supplementary Fig. 7b). Similar results were observed in an adult LRMD dog (Supplementary Fig. 7d).

**Non-Cas9 AAV vectors do not induce CTL in affected dogs**. To exclude the potential contribution of AAV capsid immunity and vector stock impurity, we made two other AAV8 vectors using identical methods. One vector expressed flag-tagged sarco/endoplasmic reticulum calcium ATPase 2a (SERCA2a) cDNA, and the other expressed flag-tagged micro-dystrophin gene (Fig. 1q, r and Supplementary Fig. 7e–g). These vectors were injected to 10-month-old affected dogs under a standard prednisolone immune suppression regime that was used in all dogs treated with non-Cas9 AAV vectors in this manuscript (Supplementary Fig. 6g). Robust SERCA2a and micro-dystrophin expression were detected for at least 14 and 84 weeks, respectively, with minimal T-cell infiltration and non-significant muscle cytokine elevation (Fig. 1q, r and Supplementary Fig. 7f, g). Consistent with the literature[25], persistent expression was associated with FoxP3+ T cells (Fig. 1q). These results suggest AAV capsid and vector preparation methods did not cause the CTL response seen in CRISPR-treated DMD dogs.

**The Cas9 AAV vector is responsible for the CTL response**. To exclude the contribution of the gRNA vector, we co-injected CK8.Cas9 and RSV.AP vectors to 1-week-old and 1-month-old normal puppies, and the CK8.Cas9 vector to normal adult dogs (Fig. 2 and Supplementary Figs. 5d and 8). As controls, age-matched normal dogs were injected with AAV.RSV.AP only. High-level persistent AP expression was detected in all controls without T-cell infiltration and muscle cytokine elevation (Fig. 2a, e, f, j, k, p). In Cas9 and AP vector co-injected puppies, AP expression was substantially reduced at 6 weeks post-injection. This reduction was again accompanied by robust CD4+ and CD8+ T-cell infiltration, vector genome loss, and muscle cytokine elevation (Fig. 2a, b, e, f, g, j). CK8.Cas9-injected adult dog muscle appeared normal at 3 weeks post-injection but showed substantial

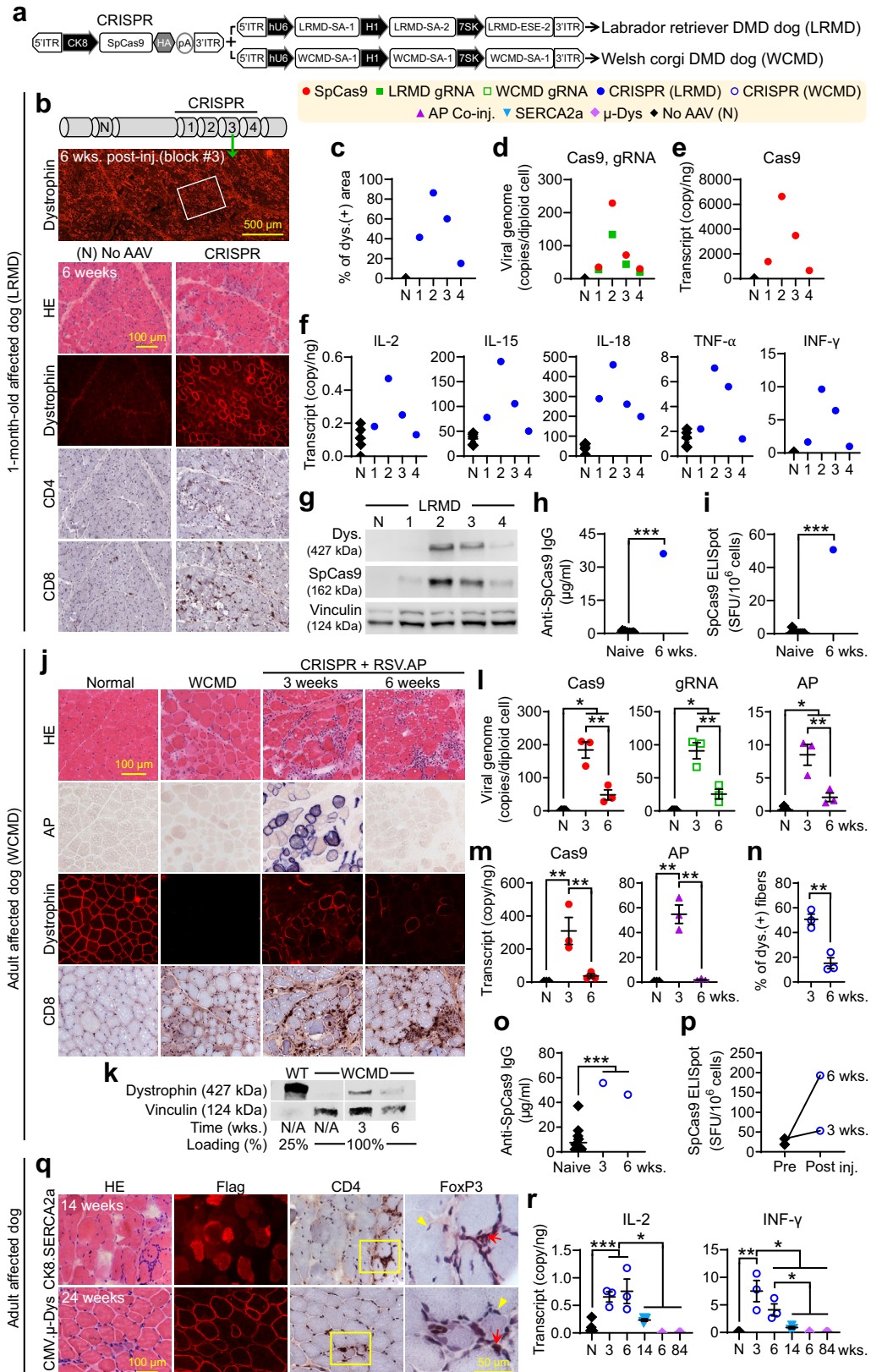

T-cell infiltration, muscle cytokine elevation, muscle cell death, and dystrophin reduction at 6 weeks post-injection (Fig. 2k–m, p and Supplementary Figs. 5d and 8a). Importantly, granzyme B⁺ T-cells were detected around dying muscle cells, suggesting active killing (Fig. 2k). Serum Cas9 antibody elevation suggested a humoral response (Fig. 2c, h, n). PBMC ELISpot was negative and

serum cytokine levels were unremarkable (Supplementary Fig. 8b, c). However, a low but significant T-cell response was detected at 6 weeks post-injection by ELISpot in draining lymph node cells (Fig. 2d, i, o), indicating a local response. Additional studies revealed a similar T-cell response to SpCas9 expressed from a ubiquitous promoter, SaCas9 expressed from the CK8 promoter,

**Fig. 1 Cytotoxic T-lymphocyte response to intramuscular AAV CRISPR therapy attenuated dystrophin restoration in canine DMD models. a** Cas9 and gRNA vectors. **b–i** AAV CRISPR therapy induced strong immune responses in the LRMD model. **b** Representative photomicrographs from non-injected and block 3 of the injected muscle. **c** Dystrophin (+) myofiber quantification. **d** Cas9 and gRNA vector genome quantification. **e** Cas9 transcript quantification (for **c**, **d**, **e**, $n = 1$ for all categories). **f** Muscle cytokine quantification (N, $n = 5$; others, $n = 1$). **g** Dystrophin and Cas9 western blot. **h** Serum Cas9 antibody. **i** Cas9-specific IFN-γ ELISpot assay on PBMCs (for **h** and **i**: Naive, $n = 11$; 6 wks., $n = 1$). **j-p** Cytotoxic T-lymphocyte response induced by AAV CRISPR therapy cleared rescued dystrophin in the WCMD model. Muscle was co-injected with an RSV.AP reporter vector. **j** Representative photomicrographs. **k** Dystrophin western blot. **l** Cas9, gRNA and AP vector genome quantification (N, $n = 5$; 3 and 6 wks., $n = 3$). **m** Cas9 and AP transcript ($n = 3$ for all categories). **n** Dystrophin (+) myofiber quantification ($n = 3$ for both). **o** Serum Cas9 antibody (Naive, $n = 14$; 3 and 6 wks., $n = 1$). **p** Cas9-specific IFN-γ ELISpot assay on PBMCs ($n = 1$ for all categories). **q** Representative photomicrographs from affected dogs that were injected with flag-tagged CK8.SERCA2a and CMV.micro-dystrophin (μDys) vectors. Red arrow, FoxP3+ T cells; Yellow arrowhead, FoxP3-negative myonuclei. **r**, Muscle interleukin-2 (IL-2) and IFN-γ transcript quantification (N, $n = 14$; $n = 3$ for each category except $n = 1$ for μDys 6 wks). Immune cells are stained in dark brown and AP is stained in blue. Numbers (1, 2, 3, 4) in **b–g** are muscle block numbers. AP, alkaline phosphatase. N, No AAV. N/A, non-applicable. wks., weeks post-injection. Data are mean ± SEM. Statistical analysis was performed using Crawford-Howell test for **h**, **i**, **o** One-way ANOVA with Tukey's multiple comparisons for **l**, **m**, and **r**, and Student's t-test for **n**. See source data file for the exact p-value. *$p < 0.05$; **$p < 0.01$; ***$p < 0.001$.

and SpCas9 expressed from the AAV9.CK8.SpCas9 vector in normal adult dogs (Supplementary Fig. 9).

**Systemic AAV CRISPR leads to wide dystrophin rescue and CTL.** While intramuscular injection is suitable for treating certain muscle diseases[26,27], others require systemic delivery for effective therapy[28]. Previous studies suggested that intravenous AAV injection may be less immunogenic[29]. To determine whether systemic delivery can minimize Cas9 immunity, we co-injected CK8.Cas9 and gRNA vectors ($1 \times 10^{14}$ vg/kg each) to two 1-month-old LRMD dogs intravenously. As a control, we injected a 2.5-month-old affected dog with $1 \times 10^{14}$ vg/kg micro-dystrophin vector intravenously (Fig. 3 and Supplementary Figs. 10–13). Vigorous micro-dystrophin expression was detected in the control dog for 88 weeks without muscle inflammation (Fig. 3o, p and Supplementary Fig. 12c). In CRISPR-treated dogs, we observed dystrophin restoration but also CD4+ and CD8+ T-cell infiltration at 3 weeks post-injection (Fig. 3a, h). One dog was harvested at 6 weeks post-injection and the other at 12 weeks post-injection. Bodywide dystrophin rescue (8–45% by immunostaining, 0.8–16.3% by immunoblots) was detected in both dogs but was associated with ample CD8+ T-cell infiltration and muscle cytokine elevation (Fig. 3b, c, i, j, p and Supplementary Figs. 10–12). Granzyme B staining confirmed active killing (Fig. 3c, j). The vector genome and Cas9 transcript were detected at 3 and 6 weeks, but substantially diminished at 12 weeks (Fig. 3d, e, k, l). Serum Cas9 antibody and PBMC ELISpot confirmed Cas9-specific responses in both dogs (Fig. 3f, g, m, n).

**Systemic AAV.Cas9 induces the immune response in normal dogs.** To determine whether AAV dose and dystrophic pathology underlay Cas9 immunity in systemic CRISPR therapy, we co-injected two 1-m-old normal puppies with RSV.AP and lower doses of CK8.Cas9 vectors intravenously (Fig. 4 and Supplementary Fig. 14). A normal control puppy received AAV.RSV.AP. Persistent AP expression was detected in the control puppy (Fig. 4m–o). However, AP expression declined rapidly in puppies co-injected with the CK8.Cas9 vector (Fig. 4a, d, g, j). Cas9 vector-injected puppies also showed concomitant loss of the vector genomes and transcripts, elevation of the serum Cas9 antibody, and increased PBMC responses by ELISpot (Fig. 4b, c, e, f, h, i, k, l). Intriguingly, there were minimal T-cell infiltrates or changes in muscle cytokines (Fig. 4a, g, p).

In summary, we demonstrated efficient AAV CRISPR-mediated dystrophin restoration by local and systemic injection in canine DMD models. These results echo well with previous reports in affected puppies and piglets[18,19]. However, by extending our studies to adult animals and including comprehensive immune

assays, we showed that Cas9-specific immune responses represent a critical barrier for AAV CRISPR therapy in large mammals. This conclusion is further supported by a recent mouse study[30]. Importantly, we show that strategies commonly used to minimize cellular immune responses (e.g., tissue-specific promoter, prednisolone immune suppression) are insufficient to bypass Cas9 immunity. Comprehensive approaches, including but not limited to the use of an immune-attenuated/immune-privileged Cas9 gene expression cassette, nonviral vector-mediated transient Cas9 gene expression, immune tolerance induction, more potent immune suppression, and pre-screening for Cas9 immunity, might be necessary in order to translate AAV CRISPR therapy to DMD patients[10,13,31].

## Methods

**Experimental dogs.** All animal experiments were conducted at the University of Missouri (except for serum collection at Auburn University described below) and approved by the Animal Care and Use Committee of the University of Missouri, were complied with all relevant ethical regulations for animal testing and research, and were performed in accordance with National Institutes of Health guidelines. Serum from a subset of dogs shown in Supplementary Fig. 6b was collected at Auburn University. This was approved by the Animal Care and Use Committee of Auburn University. All experimental dogs were on a mixed genetic background of the golden retriever, Labrador retriever, beagle, and Welsh corgi and were generated in-house by artificial insemination. All affected dogs carry null mutations in the dystrophin gene. The genotype was determined by polymerase chain reaction (PCR) according to published protocols and confirmed by the significantly elevated serum creatine kinase level[32,33]. The age, sex, and sample size are summarized in Supplementary Table 7.

All experimental dogs were housed in an AALAC accredited, limited access, conventional animal care facility and kept under a 12-h light/12-h dark cycle. Affected dogs were housed in a raised platform kennel, while normal dogs were housed in a regular floor kennel. Depending on the age and size, two or more dogs were housed together to promote socialization. Normal dogs were fed dry Purina Lab Diet 5006, while affected dogs were fed wet Purina Proplan Puppy food. Dogs were given ad libitum access to clean drinking water. Toys were allowed in the kennel with dogs for enrichment. Dogs were monitored daily by caregivers for overall health condition and activity. A complete physical examination was performed by veterinarians from the Office of Animal Research at the University of Missouri for any unusual changes in behavior, activity, food, and water consumption or when clinical symptoms were noticed. The body weights of the dogs were measured periodically to monitor growth. Anesthetized experimental subjects were euthanized according to the 2013 AVMA Guidelines for the Euthanasia of Animals.

Three canine DMD models were used to study CRISPR therapy and Cas9-induced immune responses (Supplementary Fig. 1a). The GRMD model carries a point mutation in intron 6 that disrupts the splice acceptor signal in exon 7[34]. This leads to erroneous splicing of exon 6 to exon 8. The resulting transcript is out-of-frame. The reading frame in the GRMD model can be restored by removing exons 6, 7, and 8 using the two-gRNA approach, one gRNA targeting intron 5 and the other targeting intron 8. The WCMD model contains a long interspersed repetitive element-1 (LINE-1) insertion in intron 13. This introduces a new exon carrying an in-frame stop codon[32]. The reading frame in the WCMD model can be restored by removing the new exon using a single gRNA that targets the splice acceptor (SA) and/or exonic splicing enhancer (ESE). The LRMD model carries a mutation similar to that of the WCMD model except that the repetitive element is inserted in

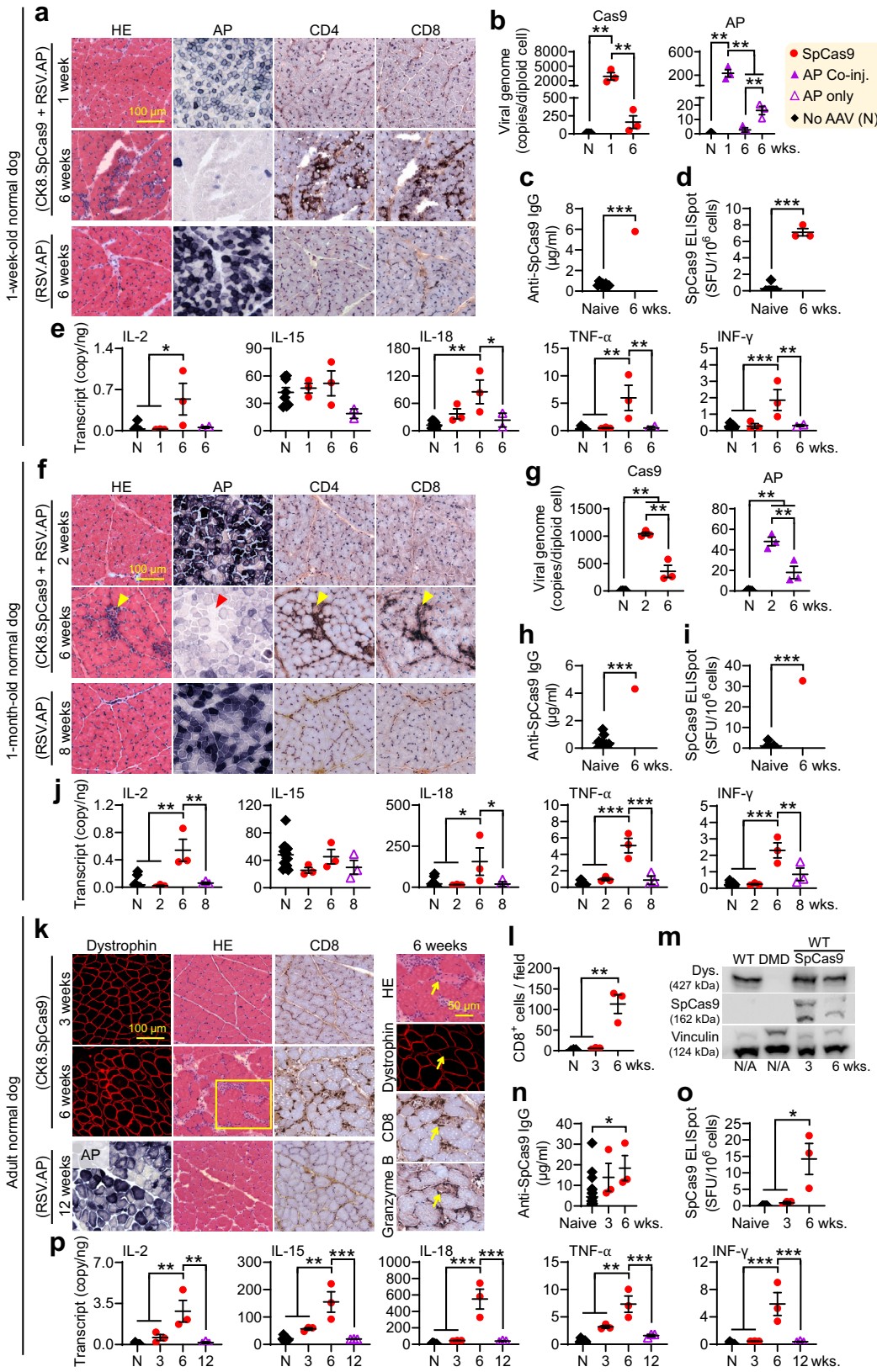

intron 19 instead of 13. A similar single gRNA approach can be used to restore dystrophin expression in the LRMD model (Supplementary Fig. 1a).

**Design of the gRNA for GRMD editing**. The gRNAs were designed to target introns 5 and 8 of the canine *dystrophin* gene (Supplementary Fig. 1a). All NNGRRT PAM targets within introns 5 and 8 were identified and then selected based on the score from the Cas-OFFfinder software (http://www.rgenome.net/cas-offinder)[35]. Top candidate gRNAs are listed in Supplementary Table 1.

**Surveyor screening for GRMD editing gRNAs**. The GRMD myoblasts were extracted from skeletal muscle according to a published protocol[36]. Myoblasts were maintained in PromoCell skeletal muscle cell growth medium (PromoCell,

**Fig. 2 AAV-mediated Cas9 expression induced humoral and cellular immune responses in normal dogs following intramuscular injection. a–e** 1-week-old normal dogs were either co-injected with AAV.CK8.SpCas9 and AAV.RSV.AP or injected with AAV.RSV.AP only. **a** Representative HE, AP, CD4, and CD8 stainings. **b** Cas9, and AP vector genome quantification (N, $n = 5$; others, $n = 3$). **c** Serum Cas9 antibody (Naive, $n = 8$; 6wks., $n = 1$). **d** Cas9 ELISpot assay on lymphocytes (Naive, $n = 5$; 6wks., $n = 3$). **e** Muscle cytokine transcript quantification (Naive, $n = 8$; others, $n = 3$). **f–j** Similar to **a–e** except 1-m-old dogs were injected. **f** Representative HE, AP, CD4, and CD8 stainings. Arrowhead, T-cell infiltration wiped out AP expression. **g** Cas9 and AP vector genome quantification (N, $n = 4$; others, $n = 3$). **h** Serum Cas9 antibody (N, $n = 12$; 6wks., $n = 1$). **i** Cas9 ELISpot assay on lymphocytes (Naive, $n = 5$; 6wks., $n = 1$). **j** Muscle cytokine transcript quantification (N, $n = 14$; others, $n = 3$). **k–p** Similar to **a–e** except adult dogs were injected with either AAV.CK8.SpCas9 or AAV.RSV.AP. **k** Representative HE, AP, CD8, and granzyme B stainings. The boxed region was magnified. Yellow arrowhead, a dying myofiber lost dystrophin expression and infiltrated with CD8$^+$ and granzyme B$^+$ T cells. **l** CD8$^+$ T-cell quantification ($n = 3$ for all categories). **m** Dystrophin and Cas9 western blot. **n** Serum Cas9 antibody (Naive, $n = 27$; others, $n = 3$). **o** Cas9 ELISpot assay on lymphocytes (Naive, $n = 6$; others, $n = 3$). **p** Muscle cytokine transcript quantification from adult dogs (N, $n = 12$; others, $n = 3$). Immune cells are stained in dark brown, and AP is stained in blue. AP, alkaline phosphatase. N, No AAV. N/A, non-applicable. wks., weeks post-injection. Data are mean ± SEM. Statistical analysis was performed using Crawford-Howell test for **c**, **h**, **i**, One-way ANOVA with Tukey's multiple comparisons for **b**, **e**, **g**, **j**, **l**, and **n-p** and Student's t-test for **d**. See source data file for the exact p-value. *$p < 0.05$; **$p < 0.01$; ***$p < 0.001$.

Heidelberg, Germany) supplemented with 50 μg/mL fetuin (Sigma-Aldrich, Saint Louis, MO, USA), 10 ng/mL human epidermal growth factor (Sigma-Aldrich), 1 ng/mL human basic fibroblast growth factor (Sigma-Aldrich), 10 μg/mL insulin (Sigma-Aldrich), 0.4 μg/mL dexamethasone (Sigma-Aldrich), 20% fetal bovine serum (Sigma-Aldrich), 1% GlutaMAX (ThermoFisher Scientific, Waltham, MA, USA), and 1% penicillin-streptomycin (ThermoFisher Scientific). Myoblasts were resuspended in calcium and magnesium-free Dulbecco's phosphate-buffered saline (ThermoFisher Scientific) and transfected with 5 μg each of the gRNA and the SaCas9 expressing plasmid by electroporation with the Gene Pulser Xcell™ electroporator (Bio-Rad, Hercules, CA, USA). After electroporation, cells were immediately immersed in the PromoCell skeletal muscle cell growth medium (PromoCell) and incubated in gelatin-coated plates. After 72 h, cells were pelleted and genomic DNA extracted with the DNeasy kit (Qiagen, Hilden, Germany). The genomic DNA was PCR amplified for 35 cycles using Acuprime HiFi Taq polymerase (ThermoFisher Scientific) and the primers listed in Supplementary Table 2. The PCR product was digested with the Surveyor enzyme (IDT) according to the manufacturer's instructions. Digested PCR products were electrophoresed on a 2% agarose gel and imaged with the Image Lab software (Version 3.0, Bio-Rad) using the GelDoc Imaging System (Bio-Rad) (Supplementary Fig. 1b and 2).

**Design of single gRNA for WCMD and LRMD mutations**. The LRMD and WCMD dogs carry repetitive element insertion in introns 19 and 13, respectively. This introduces a new exon containing an in-frame stop codon. Dystrophin expression in LRMD and WCMD dogs can be restored by blocking the splicing of the new exon using gRNAs targeting SA or ESE. Hence, the gRNAs were designed to target SA or ESE (Supplementary Fig. 1c and Supplementary Tables 3-5). The ESE's for specific exons were predicted using the ESEfinder software (Version 3.0, Cold Spring Harbor Laboratory, http://rulai.cshl.edu)[37,38]. The new exon resulted from repetitive element insertion was used as the query to predict ESEs. The SRProteins matrix library and default threshold values were used for prediction. The SA targeting gRNAs were manually designed using conserved "AG" as the splicing acceptor site in the new exon. The ESE gRNA sequences were then designed and scored using the uCRISPR algorithm[39]. Briefly, the query sequences (both plus and minus strands) were scanned to find all the possible on-target sites (20-nt) with NRG as the PAM sequence, and each (on-target) case was scored by uCRISPR algorithm. The gRNAs that have high on-target scores and specificity scores, and also cut within the ESE sequence were selected for in vitro screening. For the off-target effects, the Cas-OFFinder software[35] was used to search for all the potential genome-wide off-target sites (with no more than 4 base pair mismatches in RNA-DNA heteroduplex) from the canFam3 assembly of UCSC Genome Browser (https://genome.ucsc.edu)[40], and each off-target case was scored by the uCRISPR algorithm. Manually designed SA targeting sequences were scored using the same method.

**In vitro screening of the gRNA for LRMD and WCMD editing**. The cleavage efficiency was determined using the Guide-it™ gRNA In Vitro Transcription and Screening System (Takara Bio Inc., Mountain View, CA, USA) (Supplementary Figs. 1d and 3). Briefly, the gRNA encoding template was generated by PCR and reverse transcribed to generate the gRNA. The gRNA was then purified and quantified. In a separate reaction, genomic DNA extracted from LRMD and WCMD was PCR amplified to generate a target DNA sequence of 704 and 741 bp long, respectively (Supplementary Table 6). The purified target-specific gRNA (50 ng) was combined with 250 ng recombinant SpCas9 nuclease. About 100–250 ng of target DNA was incubated with the Cas9/gRNA mixture at 37 °C for 1 h. The cleaved product was analyzed in a 2% agarose gel. A control target DNA fragment was cleaved simultaneously with a control gRNA combined with recombinant Cas9. The band intensity was quantified by densitometry analysis using the Image Studio™ Lite Software (Version 5.2.5, LI-COR Inc., Lincoln,

Nebraska, USA). The cleavage efficiency was determined based on the band intensity.

**The gRNA cis-plasmids for the GRMD editing**. Two different gRNA cis-plasmids were generated, one without the SaCas9 gene (called pXP76, GRMD-1) and the other with the SaCas9 gene (called pXP78, GRMD-2) (Supplementary Fig. 4a). The pXP76 was generated by modifying the gRNA cis-plasmid we published before for mdx editing[17]. Briefly, the gRNA sequences for mdx editing were swapped out with the gRNA sequences for GRMD editing (CCR73 and CCR84). The pXP78 was generated by inserting the CK8-SaCas9 expression cassette between the 5′-ITR and the first U6 promoter.

**The gRNA cis-plasmids for the LRMD and WCMD editing**. The cis-plasmids were cloned using a template and intermediate assembly plasmids published before (gifts from Drs. Dirk Grimm and Eric Olson)[41]. They were named pNW29 (LRMD-4), pNW42 (LRMD-1), pNW43 (LRMD-3), and pNW55 (LRMD-2) for LRMD editing and pNW33 (WCMD-4), pNW44 (WCMD-2), pNW45 (WCMD-3) and pNW56 (WCMD-1) for WCMD editing (Supplementary Fig. 4a). The cloning of the AAV cis-plasmids was done using a BbsI (BpiI) restriction enzyme site. Two consecutive assembly steps using a Golden Gate Assembly (New England Biolabs Inc., Ipswich, MA, USA) were performed to clone the selected gRNAs to the template backbone. First, the individual gRNAs were synthesized as oligonucleotides, and 100 μM of each oligonucleotide was annealed using the IDT annealing buffer (IDT, Newark, NJ, USA) on a benchtop heat block for 5 min at 95 °C and the heat block was turned off and allowed cooling down to the ambient temperature. Annealed oligos were diluted with water to 1:200 prior to cloning. The donor plasmids carried either the U6, H1 or 7SK promoter. The diluted annealed oligos were cloned to each donor plasmid in a 10 μl reaction containing 40 fmol destination backbone, 1 μl annealed, diluted oligos, 0.75 μL of Esp3I (Thermo-Fisher Scientific), 1 μl buffer tango (Thermo-Fisher Scientific), 1 μl of T4 DNA Ligase (400 U/μl) (New England Biolabs Inc.) as well as adenosine 5′-triphosphate (New England Biolabs Inc.) and dithiothreitol (ThermoFisher Scientific) each at a final concentration of 1 mM. Cloning was performed using an Applied Biosystems thermocycler (ThermoFisher Scientific) programmed for 25 to 50 cycles of 37 °C 3 min followed by 20 °C 5 min. The restriction enzyme and ligase were denatured by heating to 80 °C for 20 min. A 3 μl volume of this reaction was transformed to Sure-2 competent cells (Agilent Technologies, Santa Clara, CA, USA), recovered in SOC medium (2% tryptone, 0.5% yeast extract, 10 mM NaCl, 2.5 mM KCl, 10 mM MgCl₂, 10 mM MgSO₄, and 20 mM glucose) and plated on Luria-Bertani agar (Sigma-Aldrich) plates containing chloramphenicol (Sigma-Aldrich) (25 μg/mL). Each plasmid was screened for correct assembly using the sequencing primer 5′-AACGGGCAAGGTGTCACCACCC-3′. The verified donor plasmids carried the gRNA under the U6, H1, or 7SK promoter. In the second step of the assembly, 20 fmol for each of the three donor plasmids, the AAV recipient plasmid, 0.75 μl BpiI (Thermo-Fisher Scientific), 2 μl buffer green (Thermo-Fisher Scientific), 0.5 μL T4 DNA Ligase (2000 U/μL) (New England Biolabs Inc.), 1 mM ATP (final concentration), and 1 mM dithiothreitol (final concentration) were mixed in a 20 μl reaction. The assembly was performed using the same thermocycler program described above. A 5 μl volume of the assembly was transformed to Sure-2 competent cells (Agilent Technologies) and spread on Luria-Bertani agar plates with ampicillin (Sigma-Aldrich). The cloning was confirmed by sequencing.

**SpCas9, SaCas9, and AP cis-plasmids**. The CK8.SaCas9 plasmid (called pXP65) was cloned by swapping the CMV promoter in our previously published pTRSaCas9[17] with the CK8 promoter[18,22,42]. The CK8.SpCas9 (called pXP116) was cloned by swapping the SaCas9 gene in pXP65 with the SpCas9 gene from the Feng Zhang Lab[43]. The CB.SpCas9 cis-plasmid was a gift from Dr. Guangping Gao[44]. An HA tag was fused to the C-terminal end of SaCas9 and SpCas9 for easy

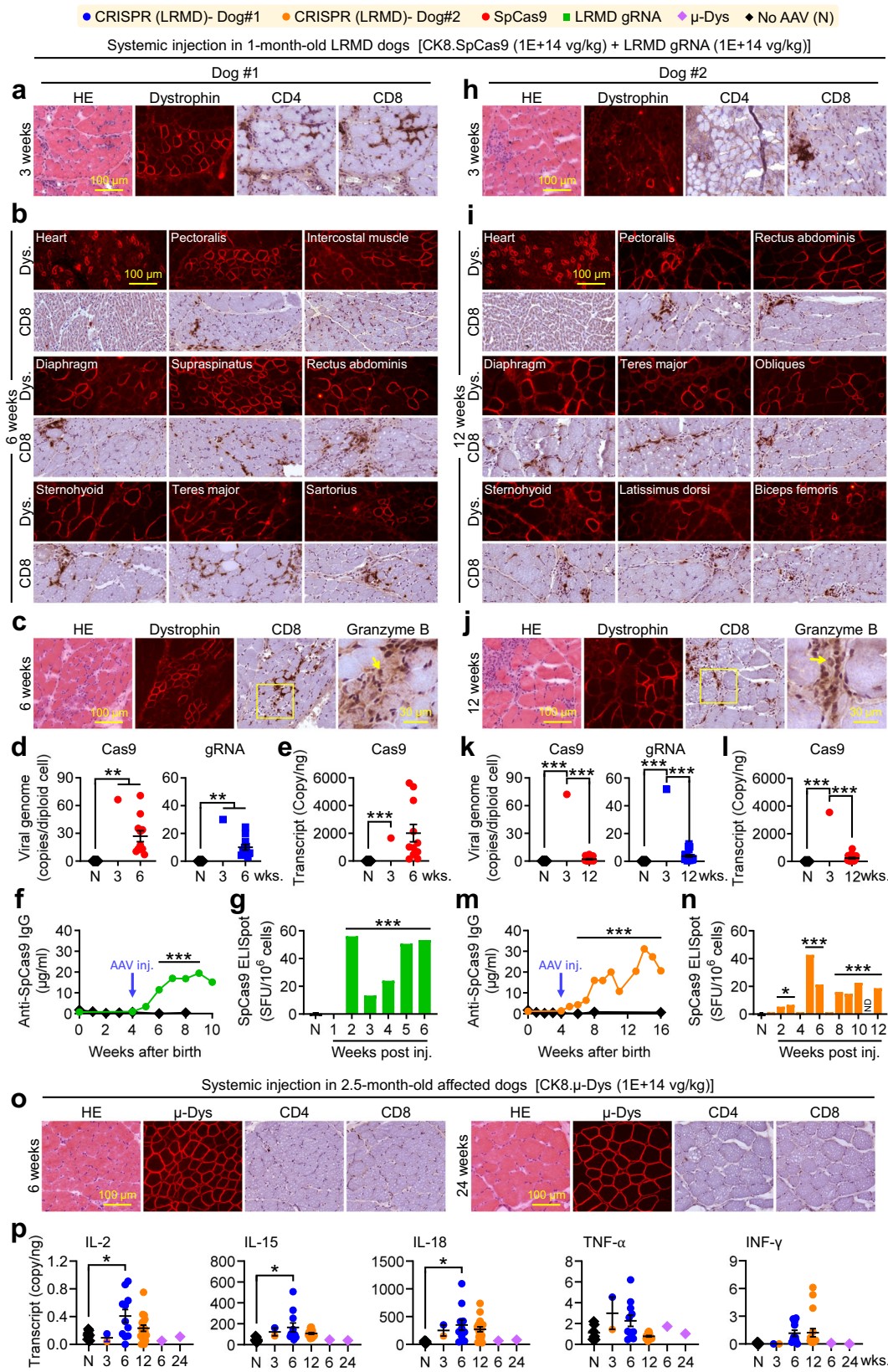

detection of the Cas9 protein by western blot. The AP cis-plasmid (pcisRSV.AP) was published previously[45]. All cis-plasmids were confirmed by sequencing.

**AAV production and purification.** The stock AAV8 and AAV9 vectors were produced using our previously reported triple plasmid transfection protocol using 293 cells[46,47]. AAV was purified through three rounds of isopycnic CsCl ultracentrifugation followed by three changes of HEPES buffer at 4 °C for 48 h. Viral titer and quality control were performed according to our previously published protocol[47]. Endotoxin contamination was examined using the Endosafe Limulus Amebocyte Lysate (LAL) gel clot test assay kit (Charles Rivers Laboratories, Wilmington, MA, USA). The endotoxin levels in our viral stocks were within the acceptable level recommended by the Food and Drug Administration. AAV purity was confirmed by silver staining[24,48].

**Fig. 3 Systemic AAV CRISPR therapy induced humoral and cellular immune responses against Cas9. a** Representative HE, dystrophin, CD4, and CD8 stainings from biopsy. **b**, Representative dystrophin and CD8 stainings from necropsied muscles. **c** Representative HE, dystrophin, CD8, and granzyme B staining from a necropsied muscle. **d** Cas9 and gRNA vector genome quantification (N, $n = 7$; 3wks., $n = 1$; 6wks., $n = 11$). **e** Cas9 transcript quantification (N, $n = 7$; 3wks., $n = 1$; 6wks., $n = 11$). **f** Serum Cas9 antibody (N, $n = 16, 8, 9, 11, 12, 23, 11$ and $18$ for $0, 1, 2, 3, 4, 6, 8$, and $10$ weeks after birth, respectively; dog#1, $n = 1$ for $0, 4$–$10$ weeks after birth). **g** PBMC IFN-$\gamma$ ELISPOT against Cas9 (N, $n = 11$; dog #1 $n = 1$ for all-time points). **h** Representative HE, dystrophin, CD4, and CD8 staining from biopsy. **i** Representative dystrophin and CD8 staining from necropsied muscles. **j** Representative HE, dystrophin, CD8, and granzyme B stainings from a necropsied muscle. **k** Cas9 and gRNA vector genome quantification (N, $n = 7$; 3wks., $n = 1$; 6wks., $n = 18$). **l** Cas9 transcript quantification (N, $n = 7$, 3wks., $n = 1$; 6wks., $n = 18$). **m** Serum Cas9 antibody (N, $n = 16, 8, 9, 11, 12, 23, 11$ and $18$ for $0, 1, 2, 3, 4, 6, 8$, and $16$ weeks after birth, respectively; dog#2, $n = 1$ for $0, 4$–$16$ weeks after birth). **n** Cas9 specific IFN-$\gamma$ ELISpot assay on PBMCs (N, $n = 11$; dog #1 $n = 1$ for all-time points). **o** Representative HE, dystrophin, CD4 and CD8 staining from the control dog that received the micro-dystrophin vector. **p** Muscle cytokine transcript quantification from all three dogs (N, $n = 7$;dog#1, 3wks. $n = 1$, 6wks. $n = 11$; dog#2, 3wks. $n = 1$, 12wks. $n = 18$; µDys injected dog, 6wks. $n = 1$, 24wks. $n = 1$). Immune cells are stained in dark brown. Arrow, a granzyme B-positive T cell. N, No AAV. wks., weeks post-injection. Data are mean ± SEM. Statistical analysis was performed using Crawford–Howell test for **d**–**g** and **k**–**n**, One-way ANOVA with Tukey's multiple comparisons for **p**. See source data file for the exact p-value. *$p < 0.05$; **$p < 0.01$; ***$p < 0.001$.

**Transient immune suppression**. Two immune suppression regimes were used in the study (Supplementary Fig. 6f, g). The five-week high-dose prednisolone immune suppression was carried out as published before with modification[18]. Specifically, prednisolone (Letco medical, Decatur, AL, USA) was flavored (FLA-VORX, Columbia, MD, USA) and administrated orally at 4 mg/kg once a day for 3 days before AAV injection and continued for 7 days after AAV injection. The dosage was reduced to 2 mg/kg once a day for 7 days, 1 mg/kg once a day for 7 days, 0.5 mg/kg once a day for 7 days, and finally, 0.5 mg/kg every other day for 7 days. The four-week standard prednisolone immune suppression was performed by administrating flavored prednisolone orally at the dose of 1 mg/kg once a day, started 3 days before AAV injection and ended four weeks after AAV injection.

**AAV administration**. All injection was performed by CHH with the assistance of YY and JT. AAV vectors were mixed thoroughly at the indicated dosage (Supplementary Table 7) before injection. For local injection, the experimental dog was sedated with a mixture of dexmedetomidine (3–12 µg/kg) and nalbuphine (0.5–1 mg/kg). Vital signs, capillary refill time, mucous membrane color, and palpebral and pedal reflexes were recorded every 15 min. The following muscles were used for intramuscular injection, including the biceps femoris muscle, cranial tibialis muscle, semitendinosus muscle, lateral gastrocnemius muscle, deltoid, and the extensor carpi ulnaris. One or multiple muscles were injected in each experimental subject. Once the muscle was identified, the body hair was shaved, and skin was scrubbed with chlorhexidine and 70% ethanol. The injected area was marked with a sterile tattoo dye (Tommy's StarBrite colors, Somers, CT, USA) on the skin. Systemic injection was performed according to our published protocol[49,50].

For in vivo gRNA vector screening studies (Supplementary Figs. 4b, c and 5a), each AAV8 gRNA vector was co-injected with an AAV8 CK8.Cas9 vector (CK8.SaCas9 for GRMD screening and CK8.SpCas9 for LRMD and WCMD screening) to one muscle in one affected dog at the dose of $5 \times 10^{12}$ vg/muscle/vector. Three adult affected dogs were used, including one GRMD dog for GRMD CRISPR vector screening, one LRMD for LRMD CRISPR vector screening, and one WCMD for WCMD CRISPR vector screening. Injected muscles were harvested at 3 weeks post-injection (Supplementary Table 7).

For the 1-month-old LRMD dog intramuscular editing study (Fig. 1b–i), we co-injected the AAV8 CK8.SpCas9 vector and the AAV8 LRMD-4 gRNA vector at the dose of $5 \times 10^{12}$ vg/muscle/vector to one muscle of a 1-month-old LRMD dog. Injected muscle was harvested at 6 weeks post-injection (Supplementary Table 7).

For the 44-month-old WCMD dog intramuscular editing study (Fig. 1j-p, r and Supplementary Fig. 7b, c, g), we co-injected the AAV8 CK8.SpCas9 vector and the AAV8 WCMD-1 gRNA vector at the dose of $5 \times 10^{12}$ vg/muscle/vector and the AAV8 RSV.AP vector at the dose of $5 \times 10^{11}$ vg/muscle/vector. The vector mixture was injected into three muscles of each dog. Injected muscles were harvested at 3 weeks post-injection in one WCMD dog and 6 weeks post-injection in the other WCMD dog (Supplementary Table 7).

For the local SERCA2a study (Fig. 1q top panel), we injected the AAV8 CK8.SERCA2a vector intramuscularly at the dose of $3 \times 10^{13}$ vg/muscle to three 11-month-old affected dogs. One muscle was injected in each dog. Injected muscle was harvested at 14 weeks post-injection (Supplementary Table 7).

For the local micro-dystrophin study (Fig. 1q bottom panel and Supplementary Fig. 7f), we injected the AAV8 CMV.micro-dystrophin vector at the dose of $1 \times 10^{12}$ vg/muscle to two 8-month-old affected dogs. In one dog, AAV was delivered to three muscles and in the other dog, AAV was delivered to four muscles. Injected muscles were harvested at 6, 12, 24, 48, and 84 weeks post-injection (one muscle per time point except for the 84-week time point. At this time point, we harvested three muscles) (Supplementary Table 7).

For the 1-week-old normal dog local injection study (Fig. 2a–e), we co-injected the AAV8 CK8.SpCas9 vector at the dose of $5 \times 10^{12}$ vg/muscle and the AAV8 RSV.AP vector at the dose of $5 \times 10^{11}$ vg/muscle to three 1-week-old normal dogs. Two muscles were injected in each dog. One injected muscle was harvested at

1 week post-injection and the other at 6 weeks post-injection (Supplementary Table 7).

For the 1-month-old normal dog local injection study (Fig. 2f–j), we co-injected the AAV8 CK8.SpCas9 vector at the dose of $5 \times 10^{12}$ vg/muscle and the AAV8 RSV.AP vector at the dose of $5 \times 10^{11}$ vg/muscle to three 1-month-old normal dogs. Two muscles were injected in each dog. One injected muscle was harvested at 2 weeks post-injection and the other at 6 weeks post-injection (Supplementary Table 7).

For the adult normal dog Cas9 local injection study (Fig. 2k–p and Supplementary Fig. 8), we injected the AAV8 CK8.SpCas9 vector at the dose of $3 \times 10^{13}$ vg/muscle to six 40-month-old normal dogs. One muscle was injected in each dog. Injected muscle was harvested at 3 ($n = 3$ dogs) and 6 weeks ($n = 3$ dogs) post-injection (Supplementary Table 7).

For AP-only local injection control studies (Fig. 2), we injected the AAV8 RSV.AP vector at the dose of $1 \times 10^{12}$ vg/muscle to three normal dogs (one 1-week-old, one 1-month-old, and one 12-month-old). One muscle was injected in each dog. Injected muscle was harvested at 6 (for the 1-week-old dog), 8 (for the 1-month-old dog), and 12 (for the 12-month-old dog) weeks post-injection (Supplementary Table 7).

For local injection comparing the CK8 and CB promoter vectors (Supplementary Fig. 9a), we injected the AAV8 CK8.SpCas9 vector at the dose of $2.3 \times 10^{13}$ vg/muscle to one muscle of a 16-month-old normal dog and the AAV8 CB.SpCas9 vector at the dose of $1.5 \times 10^{13}$ vg/muscle to one muscle of another 16-month-old normal dog. Injected muscles were harvested at 6 weeks post-injection (Supplementary Table 7).

For local AAV8 CK8.SaCas9 injection study (Supplementary Fig. 9b), we co-injected the AAV8 CK8.SaCas9 vector at the dose of $5 \times 10^{12}$ vg/muscle and the AAV8 RSV.AP vector at the dose of $2.5 \times 10^{11}$ vg/muscle to two muscles of a 15-month-old normal dog. Injected muscles were harvested at 3 and 6 weeks post-injection (Supplementary Table 7).

For local AAV9 CK8.SpCas9 injection study (Supplementary Fig. 9c), we co-injected the AAV9 CK8.SpCas9 vector at the dose of $5 \times 10^{12}$ vg/muscle and the RSV.AP vector at the dose of $5 \times 10^{11}$ vg/muscle to two muscles of a 14-month-old normal dog. Injected muscles were harvested at 3 and 6 weeks post-injection (Supplementary Table 7).

For systemic AAV8 CRISPR therapy in 1-month-old LRMD dogs (Fig. 3a–n, p, and Supplementary Figs. 10–13), we co-injected the AAV8 CK8.SpCas9 vector and the AAV8 LRMD-4 gRNA vector at the dose of $1 \times 10^{14}$ vg/kg/vector intravenously to two 1-month-old LRMD dogs. Muscle biopsy was performed at 3 weeks post-injection. One dog was euthanized at 6 weeks post-injection and the other was euthanized at 12 weeks post-injection (Supplementary Table 7).

For systemic AAV8 micro-dystrophin therapy study (Fig. 3o, p and Supplementary Fig. 12c), we injected the AAV8 CK8.micro-dystrophin vector at the dose of $1 \times 10^{14}$ vg/kg intravenously to a 2.5-month-old affected dog. Muscle biopsy was performed at 6 and 24 weeks post-injection. The dog was euthanized at 88 weeks post-injection (Supplementary Table 7).

For systemic AAV8 CK8.SpCas9 delivery in 1-month-old normal dogs (Fig. 4a–l and Supplementary Fig. 14), we co-injected the AAV8 CK8.SpCas9 vector ($3 \times 10^{13}$ vg/kg) and AAV8 RSV.AP vector ($5 \times 10^{12}$ vg/kg) intravenously to one dog, and we also co-injected the CK8.SpCas9 vector ($8 \times 10^{12}$ vg/kg) and RSV.AP vector ($5 \times 10^{12}$ vg/kg) intravenously to another dog. Muscle biopsy was performed at 3, 6 and 12 weeks post-injection (Supplementary Table 7).

For systemic AAV8 RSV.AP delivery (Fig. 4m–p), we injected the AAV8 RSV.AP vector ($5 \times 10^{13}$ vg/kg) intravenously to a 3-month-old normal dog. Muscle biopsy was performed at 6 and 52 weeks post-injection (Supplementary Table 7).

**Tissue harvest**. The muscle was carefully dissected out and snap-frozen in liquid nitrogen-cooled isopentane in the optimal cutting temperature (OCT) compound (Sakura Finetek Inc., Torrance, CA, USA) for morphological analysis.

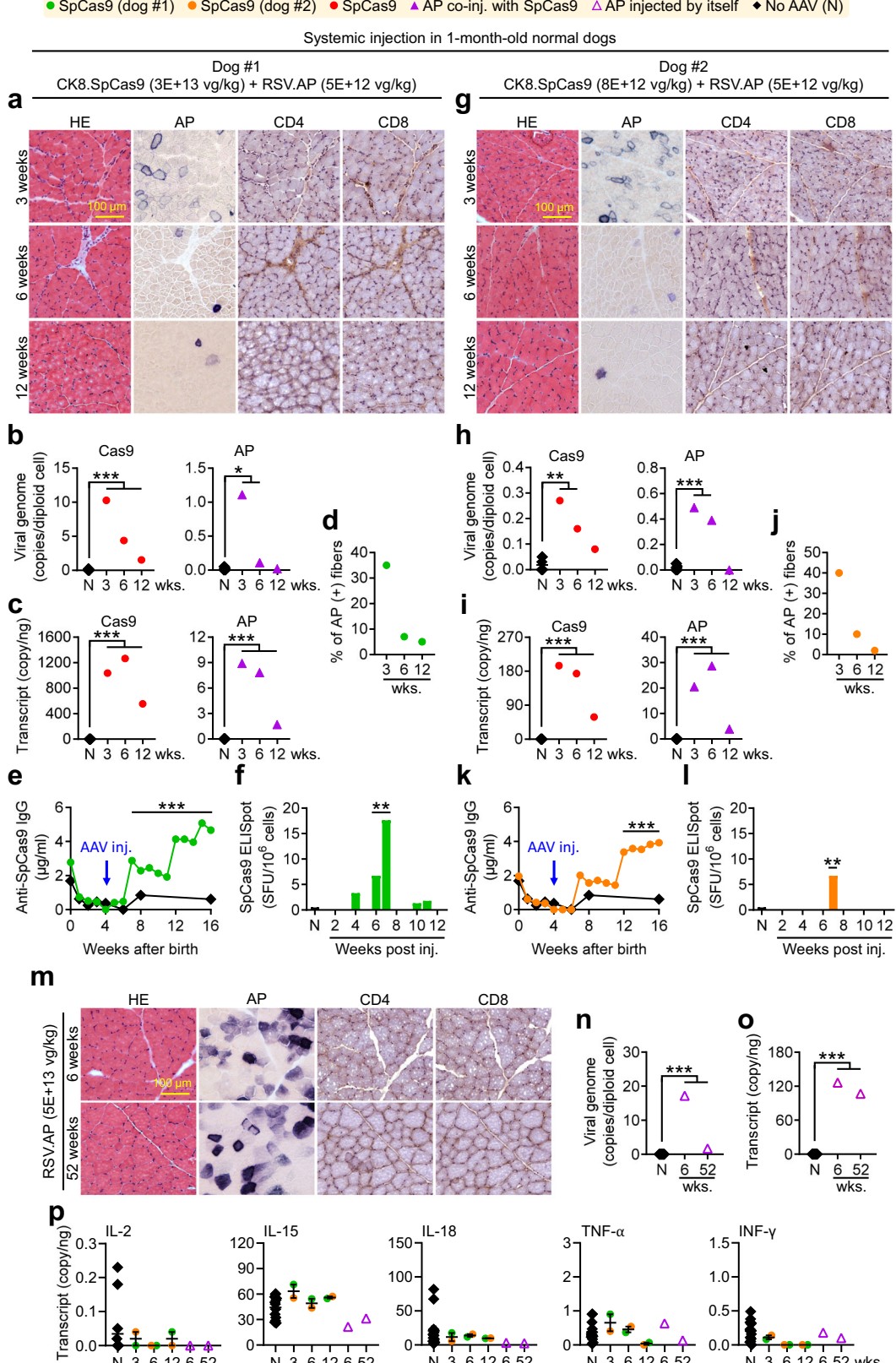

**Morphological analysis**. General muscle histopathology was revealed with hematoxylin and eosin (HE) staining. AP expression was evaluated by enzymatic histochemical staining on tissue sections according to our published protocol[51]. Briefly, samples were fixed in 0.5% glutaraldehyde for 10 min and washed in 1 mM $MgCl_2$. Endogenous AP activity was inactivated by incubating the slides at 65 °C for 45 min. The slides were then washed with a pre-staining buffer (100 mM Tris-HCl, pH 9.5, 50 mM $MgCl_2$, 100 mM NaCl) and stained in freshly prepared

staining solution (165 mg/mL 5-bromo-4-chloro-3-indolylphosphate-p-toluidine, 330 mg/mL nitroblue tetrazolium chloride, 50 mM levamisole) for 5 to 20 min.

Full-length dystrophin, edited dystrophin and micro-dystrophin were detected by immunofluorescence staining with a mouse monoclonal antibody against human dystrophin spectrin-like repeat 17 (MANEX44A, clone 5B2, 1:30, University of Iowa Hybridoma Bank, https://dshb.biology.uiowa.edu, a gift from Dr. Glen Morris)[52], and was visualized using Alexa Fluor 594 conjugated goat anti-

**Fig. 4 Systemic AAV.CK8.Cas9 injection in 1-m-old normal dogs induced humoral and cellular immune responses against Cas9. a** Representative HE, AP CD4, and CD8 stainings. **b** Cas9 and AP vector genome quantification ($N$, $n = 7$; others, $n = 1$). **c** Cas9 and AP transcript quantification ($N$, $n = 7$; others, $n = 1$). **d** AP + myofiber quantification ($n = 1$ in all categories). **e** Serum Cas9 antibody ($N$, $n = 16$, 8, 9, 11, 12, 23, 11 and 18 for 0, 1, 2, 3, 4, 6, 8, and 16 weeks after birth, respectively; dog#1, $n = 1$ for all-time points). **f** Cas9-specific IFN-γ ELISpot assay on PBMCs ($N$, $n = 11$; dog #1 $n = 1$ for all-time points). **g** Representative HE, AP, CD4, and CD8 staining. **h** Cas9 and AP vector genome quantification ($N$, $n = 7$; others, $n = 1$). **i** Cas9 and AP transcript quantification (in both panels: $N$, $n = 7$; each other group, $n = 1$). **j** AP + myofiber quantification ($n = 1$ in each group). **k** Serum Cas9 antibody ($N$, $n = 16$, 8, 9, 11, 12, 23, 11 and 18 for 0, 1, 2, 3, 4, 6, 8, and 16 weeks after birth, respectively; dog#2, $n = 1$ for all time points). **l** PBMC IFN-γ ELISpot against Cas9 ($N$, $n = 11$; dog #1 $n = 1$ for all time points). **m** Representative HE, AP, CD4, and CD8 stainings. **n** AP vector genome quantification ($N$, $n = 7$; others, $n = 1$). **o** AP transcript quantification ($N$, $n = 7$; others, $n = 1$). **p** Muscle cytokine transcript quantification from all 3 dogs at the indicated time points post-injection ($N$, $n = 14$; dog#1, 3wks. $n = 1$. 6wks. $n = 1$. 12wks. $n = 1$; dog#2, 3wks. $n = 1$. 6wks. $n = 1$. 12wks. $n = 1$; dog injected with AP vector only, 6wks. $n = 1$. 52 wks. $n = 1$). AP, alkaline phosphatase. N, No AAV. wks., weeks post-injection. Data are mean ± SEM. Statistical analysis was performed using Crawford-Howell test for **b**, **c**, **e**, **f**, **h**, **i**, **k**, **l**, **n**, **o**, and One-way ANOVA with Tukey's multiple comparisons for **p**. See source data file for exact $p$-value. *$p < 0.05$; **$p < 0.01$; ***$p < 0.001$.

mouse IgG(H + L) antibody (A11020, Lot# 1946335, 1:100, ThermoFisher Scientific).

Flag-tagged SERCA2a was detected by immunofluorescence staining with mouse monoclonal antibody against the Flag peptide (Catalog# F1804, Lot# SLBB7188, 1:500, MilliporeSigma, Burlington, MA, USA), and was visualized using Alexa Fluor 594 conjugated goat anti-mouse IgG(H + L) antibody (A11020, Lot# 1946335, 1:100, ThermoFisher Scientific).

CD4$^+$ T, CD8$^+$ T, MHCII, granzyme B, and FoxP3 were detected by immunohistochemistry staining. Rat monoclonal IgG2a antibody against canine CD4 (Catalog# MCA1038GA, clone YKIX 302.9, Lot # 1707, 1:1000, Bio-Rad Laboratories, Inc.) was used to detect CD4$^+$ T and was visualized using a biotin-conjugated goat anti-rat IgG (H + L) antibody (A10517, Lot # 1441195, 1:1000, ThermoFisher Scientific). Rat monoclonal IgG1 antibody against canine CD8 (Catalog# MCA1039, clone YCATE 55.9, Lot # 149349, 1:200, Bio-Rad Laboratories, Inc.) was used to detect CD8$^+$ T and was visualized using a biotin-conjugated goat anti-rat IgG (H + L) antibody (A10517, Lot # 1441195, 1:1000, ThermoFisher Scientific). Mouse monoclonal IgG1 antibody against canine MHC Class II Monomorphic (Catalog# MCA2037S, clone CA2.1C12, Lot # 151735, 1:100, Bio-Rad Laboratories, Inc.) was used to detect MHCII and was visualized using a biotin-conjugated goat anti-mouse IgG1 antibody (A10519, Lot # 2115665, 1:1000, ThermoFisher Scientific). Rabbit polyclonal antibody against human granzymeB (Catalog# E2580, Lot # GR3313356-1, 1:50, Spring Bioscience, Pleasanton, CA, USA) was used to detect granzymeB and was visualized using biotin-conjugated goat anti-rabbit polyclonal antibody (B8895, Lot # 086M4770V, 1:500, MilliporeSigma). Rat monoclonal IgG2a antibody against mouse FoxP3 (14–5773, clone FJK-16s, Lot # 2023701, 1:100, Thermo Fisher Scientific) was used to detect FoxP3 and was visualized using a biotin-conjugated goat anti-rat IgG (H + L) antibody (A10517, Lot # 1441195, 1:1000, ThermoFisher Scientific).

All Photomicrographs were taken with a Lecia DFC700 color camera (Leica Camera Inc., Allendale, NJ, USA) connected to a Nikon E800 microscope (Nikon Inc., Melville, NY, USA), using the Leica Application Suite software (Version 4.12.0, Leica Camera Inc.). The percentage of dystrophin positive cells was quantified from digitalized dystrophin immunostaining images using the ImageJ software (Version 1.48b, https://imagej.nih.gov). The total number of CD8$^+$ cells/filed was quantified from five random microscopic fields (200 X magnification) using the ImageJ software (Version 1.48b).

**Western blot**. Whole muscle lysate was loaded on a 6% sodium dodecyl sulfate-polyacrylamide gel, and protein was transferred to a polyvinylidene difluoride membrane. Dystrophin was detected with a rabbit polyclonal antibody against the C-terminal domain of the dystrophin protein (primary antibody) (Catalog RB-9024-P0, Lot# 9013p-1610E, 1:250, ThermoFisher Scientific) and a peroxidase-conjugated goat anti-rabbit IgG antibody (secondary antibody) (AP132P, Lot# 3123491, 1:10,000, MilliporeSigma). Cas9 was detected with a rabbit anti-HA tag monoclonal antibody (primary antibody) (Catalog 3724 S, Lot# 9, 1:500, Cell Signaling Technology, Inc., Danvers, MA, USA) and a peroxidase-conjugated goat anti-rabbit IgG antibody (secondary antibody) (AP132P, Lot# 3123491, 1:10,000, MilliporeSigma). Vinculin was used as a loading control and was detected with a mouse IgG1 monoclonal antibody (primary antibody) (V9131, clone hVIN-1, ascites fluid, Lot # 036M4797V, 1:1000, MilliporeSigma) and a peroxidase-conjugated goat anti-mouse IgG (H + L) antibody (secondary antibody) (AP124P, lot: 3032923, 1:10,000, MilliporeSigma). Signal was visualized with the LI-COR Odyssey imaging system (Li-COR Inc.) using the Image Studio™ Lite Software (Version 5.2.5, LI-COR Inc.). The full-length dystrophin protein migrated at 427 kDa and vinculin at 124 KDa. Uncropped western blot images for the entire paper are shown in Supplementary Figs. 5 and 11. Densitometry quantification was performed using the Image Studio™ Lite Software (Version 5.2.5, LI-COR Inc.). The relative intensity of the protein was normalized to the corresponding vinculin band in the same blot and further normalized to the normal control in the same blot (Supplementary Figs. 10b, d and 11).

**Vector genome copy number quantification**. Genomic DNA was extracted from OCT-embedded tissues. DNA concentration was determined by the Qubit dsDNA high-sensitivity assay kit (ThermoFisher Scientific) using the Qubit 3.0 Fluorometer (ThermoFisher Scientific). The vector genome copy number was quantified by quantitative PCR (qPCR) using the TaqMan Universal PCR master mix (ThermoFisher Scientific) and custom-designed TaqMan primers and probes (Supplemental Table 8). The qPCR was performed in the ABI 7900HT qPCR machine (ThermoFisher Scientific) using the Sequence Detection System (SDS) software (Version 2.4, ThermoFisher Scientific). The threshold cycle (Ct) value of each reaction was determined using the SDS software (Version 2.4, ThermoFisher Scientific). The vector genome copy number was determined by the Ct value that was first converted to the total copy number in the reaction using a standard curve of the known amount of the cis-plasmid, and then divided by the amount of diploid genome present in the total ng of genomic DNA used in the reaction.

**Vector transcript quantification**. RNA was extracted from OCT-embedded tissues using the RNeasy Fibrous Tissue kit (Qiagen). The cDNA was generated using the Super-script IV Kit (ThermoFisher Scientific) and quantified by the Qubit ssDNA assay kit (ThermoFisher Scientific) using the Qubit 3.0 Fluorometer (ThemoFisher Scientific). The Cas9 and AP transcripts were quantified by qPCR using the TaqMan Universal PCR master mix (ThermoFisher Scientific) and TaqMan custom-designed primers and probes (Supplemental Table 8). The qPCR was performed in the ABI 7900HT qPCR machine (ThermoFisher Scientific) using the SDS software (Version 2.4, ThermoFisher Scientific). The transcript copy number was determined by the cycle value from a quantitative reverse transcription PCR (RT-qPCR) reaction that was first converted to the raw copy number using a standard curve of the known amount of the cis-plasmid, and then divided by the total amount of cDNA (ng) used in the reaction.

**Cytokine transcript quantification**. RNA was extracted from OCT-embedded tissues using the RNeasy Fibrous Tissue kit (Qiagen). The cDNA was generated using the SuperScript IV Kit (ThermoFisher Scientific) and quantified by the Qubit ssDNA assay kit (ThermoFisher Scientific) using the Qubit 3.0 Fluorometer (ThemoFisher Scientific). The cytokine transcript was quantified by digital droplet PCR (ddPCR) using ddPCR Supermix for Probes (no dUTP) (Bio-Rad Laboratories, Inc.) and custom-designed TaqMan primers and probes (Supplemental Table 8). The ddPCR was performed in the QX200 ddPCR system (Bio-Rad Laboratories, Inc.) using the QuantaSoft software (Version 1.0, Bio-Rad). The data were analyzed with the QuantaSoft software (Version 1.0, Bio-Rad Laboratories, Inc.) and reported as the transcript copy number per ng of cDNA used in the reaction.

**Anti-SpCas9 antibody assay**. Serum anti-SpCas9 antibody levels were evaluated blindly using an enzyme-linked immunosorbent assay (ELISA). Briefly, 25 ng/ml of the SpCas9 protein (OriGene Technologies Inc., Rockville, MD, USA) in carbonate buffer was coated on a microtiter plate. The serum was used at 1:50 dilution. Horseradish peroxidase (HRP) conjugated goat anti-dog IgG was used (1:10000; Rockland Immunochemicals, PA, USA) to detect anti-Cas9 IgG. Serial dilutions of dog IgG whole molecule (Rockland Immunochemicals Inc., Gilbertsville, PA, USA) were used to generate a standard curve. The absorbance of each standard and sample were measured at 492 nm in the EPOCH2 Microplate Spectrophotometer (BioTeck, Winooski, Vermont, USA) using the Microplate Gen 5 software (Version 3.04, BioTeck). A five-parameter standard curve fit was generated, and total IgG concentrations were calculated accordingly using the Gen5 software (Version 3.04, BioTeck). The anti-SpCas9 antibody concentration was determined by converting the reaction color absorbance value to µg/ml using a standard curve.

**SpCas9 enzyme linked immune absorbent spot (ELISpot) assay**. Frequencies of SpCas9 specific T cells were determined blindly using the canine IFN-γ ELISpot assay. PBMCs (collected at different time points in alive animals), or lymphocytes

from the draining lymph nodes (collected at the end of the experiment) were stimulated in vitro for 48 h with 10 mg/ml of SpCas9 protein (OriGene Technologies Inc.). IFN-γ producing cells were indirectly detected as spots using the canine-specific IFN-γ ELISpot kit (R&D Systems Inc. Minneapolis, MN, USA) as per the manufacturer's instructions. Spots were analyzed and counted using the ImmunoSpot software (Version 7, Cellular Technology Limited, Cleaveland, OH, USA) on the ImmunoSpot® S6 Analyzer (Cellular Technology Limited). The number of cells producing IFN-γ was presented as spot forming unit (SFU) per million cells. PMA was used as the positive control in each assay. Media was used as the negative control in each assay. Untreated naïve dogs were evaluated as the baseline. Assays were performed in triplicates. No spot was detected in all negative controls (Supplementary Fig. 15).

**Serum cytokine quantification**. Serum concentrations of 12 cytokines were measured using the Milliplex MAP Canine Cytokine/Chemokine Magnetic Bead Panel (CCYTMG-90 K-PX13, MilliporeSigma, Burlington, MA, USA) according to the manufacturer's instructions (Supplementary Figs. 8b, 13 and 14). All samples, standards, and quality controls were analyzed in duplicate in the MAGPIX reader (Luminex Corp., Austin, TX, USA) using the Luminex XPONENT software (Version 4.2, Luminex Corp.). Data were analyzed using the Belysa software (Version 1.0.19, MilliporeSigma).

**Blood chemistry**. Blood was drawn from experimental subjects before the start of immune suppression (baseline data) and periodically throughout the experiment (Supplementary Tables 9-13). The laboratory biochemical test was performed at the Veterinary Medical Diagnostic Laboratory in the University of Missouri Veterinary Medical Teaching Hospital (Columbia, MO, USA).

**Statistics and reproducibility**. All data are biological replicates. Data are presented as mean ± standard error of mean (SEM). One-way ANOVA with Tukey's multiple comparisons was performed for statistical analysis for more than two group comparisons using GraphPad PRISM software (Version 8, GraphPad Software Inc., La Jolla, California). To compare the statistical significance between two groups, the two-tailed, unpaired Student's t-test was used for parametric data, and the Mann–Whitney test was used for non-parametric data. For studies involving a single treated subject, the statistical analysis was performed with the Crawford–Howell test[53–55] using the Matlab software (Version R2020a, MathWorks, Natick, MA, USA) with the Statistics and Machine Learning Toolbox (Version 11.7, MathWorks). Please note, the Crawford–Howell test cannot be used to infer any group effects. The difference was considered significant when $p < 0.05$ (*); $p < 0.01$ (**); $p < 0.001$ (***).

**Reporting summary**. Further information on research design is available in the Nature Research Reporting Summary linked to this article.

## Data availability

The data that support the findings of this study are available from the corresponding author upon reasonable request. All the raw data in this study are provided in the figures with the exception of qPCR, ddPCR, and the anti-Cas9 antibody assay raw data. The unprocessed gels and blots are provided in Supplementary Figs. 2, 3, 5 and 11. The source data for Fig. 1c–i, l–p, and r; Fig. 2b–e, g–j, l, and n–p; Fig. 3d–g, k–n, and p; Fig. 4b–f, h–l, and n-p; Supplementary Fig. 6a–e; Supplementary Fig. 7b and g; Supplementary Fig. 8b and c; Supplementary Figs. 13 and 14 are provided with the paper.

## Code availability

The Matlab code used to analyze the statistical difference using the Crawford–Howell test is available in the Zenodo database at https://doi.org/10.5281/zenodo.5562398.

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

## Acknowledgements

This work was supported by grants from the National Institutes of Health (AR-70517 and NS-90634 to D.D.; AR-69085 to C.A.G. and D.D.; UG3-AR075336, U01-EB028901, and U01-AI146356 to CAG; GM-134919 and GM-117059 to S.-J.C.; R01AI051390 and R01 HL131093 to R.W.H.; and Intramural/Extramural research program of the National Center for Advancing Translational Sciences to CHH and NNY, Department of Defense (MD15-1-0469 to CAG and MD150133 to D.D. and C.A.G.), Hope for Javier (to D.D.), Jackson Freel DMD Research Fund (to D.D.), Parent Project Muscular Dystrophy (to D.D.), Team Joseph (to D.D.), Charley's Fund (to D.D.), Zubin's Wish (to D.D.), Jett Foundation (to DD), Cure Duchenne (to D.D.) and Destroy Duchenne (to D.D.), Muscular Dystrophy Association (MDA277360 to C.A.G.), the Duke Coulter Translational Partnership (to C.A.G.). C.E.N. was supported by a Hartwell Foundation Postdoctoral Fellowship. DOP-L was supported by the University of Missouri Life Science Fellowship and NIH training grant T32 GM008396. E.D.M. was supported by the University of Missouri Life Science Fellowship.

We thank Dr. Eric Olson for sharing the donor plasmids and AAV backbone plasmid for gRNA vector construction. We thank Drs. Steve Hauschka and Jeff Chamberlain for sharing the CK8 promoter. We thank Dr. Guangping Gao for sharing the CB.SpCas9 plasmid. We thank Dr. Glenn Morris for sharing the dystrophin antibody MANEX44A. We thank Dr. Jim Wilson for providing the AAV8 packaging plasmid. We thank Dr. Leonela Amoasii for the help with the cloning of the gRNA cis-plasmids for LRMD and WCMD editing. We thank Drs. Dawn Cornelison and Kasun Kodippili for the help with myoblast preparation. We thank Dr. Hsiao H. Yang, Mr. Alex Hinken, Mr. Matt Burke, and Ms. Cheryl Rojas for technical assistance. We thank Dr. John Dodam for the help with anesthesia. We thank Jessica Folds, Rebecca Nance, Julianna Sabol, Leilani Castaner, and Mindie Roush for the help with serum collection. We thank Cheryl Rojas and Dr. Angela Royal for their help with clinical pathology. We thank Drs. Martin Katz and James Cook for sharing canine serum samples. We thank Drs. Jeffery Henegar, Scott Korte, and Erin O'Connor at the University of Missouri Office of Animal Resources/ Animal Care Quality Assurance for their help with animal care. We thank the Bond Life Sciences Center at the University of Missouri for using the ddPCR machine.

## Author contributions

Conceived the study: C.H.H. and D.D. Designed experiments: C.H.H., S.R.P.K., D.O.P.-L., N.B.W., D.Z., C.E.N., C.A.G., S.C., R.W.H., and D.D. Performed the experiments: C.H.H., S.R.P.K., D.O.P.-L., N.B.W., D.Z., Y.Y., J.T., X.P., K.Z., E.D.M., C.E.N., S.M., J.H., J.T., J.A.L. and F.F. Analyzed the data: C.H.H., S.R.P.K., D.O.P.-L., N.B.W., D.Z., Y.Y., X.P., E.D.M., C.E.N., F.F., G.Y., N.N.Y., C.A.G., S.-J.C., R.W.H., and D.D. Provided critical reagents: F.S., D.G., and B.F.S. Wrote the paper: C.H.H. and D.D.; N.B.W. and S.-J.C. contributed to method section writing. All authors edited the paper and approved the submission.

## Competing interests

D.D. is a member of the scientific advisory board for Solid Biosciences and equity holders of Solid Biosciences. The Duan lab has received research supports unrelated to this project from Solid Biosciences and Edgewise in the last three years. RWH has received research support from Luye R&D, royalty payments from Takeda Pharmaceuticals, and consulting fees from AGTC. None of these are related to this project. C.A.G., C.E.N., C.H.H., D.D., N.B.W., Y.Y. have filed patent applications related to genome editing for Duchenne muscular dystrophy. C.A.G. is an advisor to and receives research support from Sarepta Therapeutics, Inc., is a founder of M2X2 Therapeutics, Locus Biosciences, and Element Genomics, and is an advisor to Iveric Bio and Levo Therapeutics. D.G. is a co-founder and shareholder of AaviGen GmbH. The remaining authors declare no competing interests.
