## [Peer Review File · Nature Communications]

Reviewers' Comments:

Reviewer #1:

Remarks to the Author:

The authors have performed systemic delivery and ELISpot experiments according to previous review comments, and rewrote the manuscript. However, in many cases, the quantification relied on a single data point (eg. Figs. 1h, 1i, 1o,1r, 2c, 2h, 2i, 3d, 3e, 3k, 3l, 4b, 4c, 4d, 4h, 4i, 4j, 4n, 4o, and some supplementary data). This reviewer understands the challenges associated with AAV studies in large animals, however, the statistics with one data point is not meaningful. What statistical methods were used to derive the P values of statistical significance with sample size of 1 as shown in the aforementioned figures?

Reviewer #2:

Remarks to the Author:

The manuscript now titled "Local and systemic AAV CRISPR therapy induces Cas9-specific immune responses in the canine model of Duchenne muscular dystrophy" by Hakim et al elegantly describes the immune response to cas9 protein after intramuscular and systemic administration of AAV vector encoding Cas9 and gRNA in both healthy and dystrophic dog models. The authors show that loss of dystrophin expression occurred with CRISPR gene editing but not in muscle restored through microdystrophin expression. This work clearly reports that loss of dystrophin expression is accompanied by T-cell infiltration and elevation of key cytokines that was not mitigated through standard prednisolone immunosuppression protocols. This study addresses all previous concerns and is highly suitable for publication in the current form.

Reviewer #3:

Remarks to the Author:

The authors have addressed all of my previous concerns. This is a highly significant piece of work that is likely to have an important impact in the field. I have no further concerns and am enthusiastic about the work.

Manuscript #: NCOMMS-21-15942-T

Point-by-point response

Comments are highlighted in blue color

Response to concerns raised by the Reviewer 1

The authors have performed systemic delivery and ELISpot experiments according to previous review comments, and rewrote the manuscript. However, in many cases, the quantification relied on a single data point (eg. Figs. 1h, 1i, 1o,1r, 2c, 2h, 2i, 3d, 3e, 3k, 3l, 4b, 4c, 4d, 4h, 4i, 4j, 4n, 4o, and some supplementary data). This reviewer understands the challenges associated with AAV studies in large animals, however, the statistics with one data point is not meaningful. What statistical methods were used to derive the P values of statistical significance with sample size of 1 as shown in the aforementioned figures?

Response. We thank the reviewer for the comments. We truly appreciate the reviewer's understanding of the challenges we face in large animal studies where we often must deal with cases of a single treated subject. We agree with the reviewer that a single subject certainly cannot be used to infer any group effects. To determine whether the results from one individual is statistically different from a modestly sized control group, we used the Crawford-Howell test (Crawford and Howell, 1998;Crawford and Garthwaite, 2006;Crawford et al., 2006). This test can infer whether a measure in an individual is significantly different from a control group. We explicitly stated this in the online content, materials and methods section, under statistical analysis. Specifically, we stated "For studies involving a single treated subject, the statistical analysis was performed with the Crawford-Howell test ²¹⁻²³ using the Matlab software (version R2020a, Statistics and Machine Learning Toolbox 11.7)."

References

- Crawford, J.R., and Garthwaite, P.H. (2006). Methods of testing for a deficit in single-case studies: Evaluation of statistical power by Monte Carlo simulation. *Cognitive Neuropsychology* 23, 877-904.
- Crawford, J.R., Garthwaite, P.H., Azzalini, A., Howell, D.C., and Laws, K.R. (2006). Testing for a deficit in single-case studies: Effects of departures from normality. *Neuropsychologia* 44, 666-677.
- Crawford, J.R., and Howell, D.C. (1998). Comparing an individual's test score against norms derived from small samples. *Clinical Neuropsychologist* 12, 482-486.